# Mackerel and Seaweed Burger as a Functional Product for Brain and Cognitive Aging Prevention

**DOI:** 10.3390/foods13091332

**Published:** 2024-04-26

**Authors:** Carlos Cardoso, Jorge Valentim, Romina Gomes, Joana Matos, Andreia Rego, Inês Coelho, Inês Delgado, Carla Motta, Isabel Castanheira, José A. M. Prates, Narcisa M. Bandarra, Cláudia Afonso

**Affiliations:** 1CIIMAR, Interdisciplinary Centre of Marine and Environmental Research, University of Porto, Rua dos Bragas 289, 4050-123 Porto, Portugal; narcisa@ipma.pt (N.M.B.); cafonso@ipma.pt (C.A.); 2Division of Aquaculture, Upgrading, and Bioprospection (DivAV), Portuguese Institute for the Sea and Atmosphere (IPMA, IP), Avenida Alfredo Magalhães Ramalho, 6, 1495-165 Algés, Portugal; jlcvim@gmail.com (J.V.); rominamgomes@hotmail.com (R.G.); joanamatos7@outlook.pt (J.M.); 3Faculty of Science, University of Lisbon, Campo Grande, 1749-016 Lisbon, Portugal; 4MEtRICs/DCTB/NOVA, School of Science and Technology, NOVA University Lisbon, Caparica Campus, 2829-516 Almada, Portugal; 5Food and Nutrition Department, National Health Institute Doutor Ricardo Jorge (INSA, IP), Av. Padre Cruz, 1649-016 Lisbon, Portugal; andreia.rego@insa.min-saude.pt (A.R.); ines.coelho@insa.min-saude.pt (I.C.); ines.delgado@insa.min-saude.pt (I.D.); carla.motta@insa.min-saude.pt (C.M.); isabel.castanheira@insa.min-saude.pt (I.C.); 6Centro de Investigação Interdisciplinar em Sanidade Animal (CIISA), Faculdade de Medicina Veterinária, Universidade de Lisboa, 1300-477 Lisbon, Portugal; japrates@fmv.ulisboa.pt; 7Laboratório Associado para Ciência Animal e Veterinária (AL4AnimalS), Faculdade de Medicina Veterinária, Universidade de Lisboa, 1300-477 Lisbon, Portugal

**Keywords:** aging, algae, bioavailability, fatty acids, fish and fish products

## Abstract

Most world countries are experiencing a remarkable aging process. Meanwhile, 50 million people are affected by Alzheimer’s disease (AD) and related dementia and there is an increasing trend in the incidence of these major health problems. In order to address these, the increasing evidence suggesting the protective effect of dietary interventions against cognitive decline during aging may suggest a response to this challenge. There are nutrients with a neuroprotective effect. However, Western diets are poor in healthy n-3 polyunsaturated fatty acids (n-3 PUFAs), such as docosahexaenoic acid (DHA), iodine (I), and other nutrients that may protect against cognitive aging. Given DHA richness in chub mackerel (*Scomber colias*), high vitamin B9 levels in quinoa (*Chenopodium quinoa*), and I abundance in the seaweed *Saccorhiza polyschides*, a functional hamburger rich in these nutrients by using these ingredients was developed and its formulation was optimized in preliminary testing. The effects of culinary treatment (steaming, roasting, and grilling vs. raw) and digestion on bioaccessibility were evaluated. The hamburgers had high levels of n-3 PUFAs in the range of 42.0–46.4% and low levels of n-6 PUFAs (6.6–6.9%), resulting in high n-3/n-6 ratios (>6). Bioaccessibility studies showed that the hamburgers could provide the daily requirements of eicosapentaenoic acid (EPA) + DHA with 19.6 g raw, 18.6 g steamed, 18.9 g roasted, or 15.1 g grilled hamburgers. Polyphenol enrichment by the seaweed and antioxidant activity were limited. The hamburgers contained high levels of Se and I at 48–61 μg/100 g ww and 221–255 μg/100 g ww, respectively. Selenium (Se) and I bioaccessibility levels were 70–85% and 57–70%, respectively, which can be considered high levels. Nonetheless, for reaching dietary requirements, considering the influence of culinary treatment and bioaccessibility, 152.2–184.2 g would be necessary to ensure daily Se requirements and 92.0–118.1 g for I needs.

## 1. Introduction

Worldwide, due to increased life expectancy, dementia prevalence may increase by 2050 [1]. There are studies that support a link between diet and the onset and progression of Alzheimer’s disease (AD) [2,3]. Combined with the prevention of AD risk factors, such as heart disease, there is relevant evidence indicating diet as a preventive measure for AD [3]. Meta-analyses show little evidence for the treatment of this disease through diet, but for prevention, the evidence is more consistent [2]. Results are more promising in milder dementia, called moderate cognitive decline. Hence, food can prevent disease and, possibly, in a more efficient way than current drugs, given the beneficial synergies between nutrients in food [2].

With regard to diet and neuronal health, animal studies suggest that a high intake of docosahexaenoic acid (DHA, 22:6n-3) and polyunsaturated fatty acids (n-3 PUFAs) in general may protect against neuronal disease [4]. Several prospective observational studies point to a protective effect of a higher intake of DHA (>250 mg/day) against AD risk [5]. Thus, prevention is more effective than treatment [5]. Additionally, vitamin B12 (and B9) has been shown to be relevant for AD prevention. Such vitamins decrease homocysteine levels, preventing the accumulation of beta-amyloid peptides in neurons [6]. Homocysteine is well known to be a modifiable risk factor for cognitive decline; its concentration is higher in AD patients. Regarding the elemental components of diet, the prophylactic role of Se intake in AD has been suggested [7]. Primary prevention should aim at an adequate Se intake, thereby securing the optimal expression of selenoproteins. Song et al. (2018) [8] showed that Se-enriched yeast inhibited β-amyloid production and modulated autophagy in a transgenic mouse model of AD. As for I, it is deemed a relevant element for healthy brain function [9]. Many brain structures seem to be affected by I deficiency, including areas such as the hippocampus, microstructures such as myelin, and neurotransmitters [9]. Therefore, it makes sense to test the potential synergies between DHA, other n-3 FAs, Se, and I [10].

In order to use such synergies, a functional food rich in DHA, Se, and I in the diet of middle-aged and elderly individuals would be advantageous. This could be facilitated by using classic food concepts with a high potential of successfully incorporating novel ingredients in their formulation—for instance, a hamburger. The ingredients should be rich sources of the aforementioned compounds. For DHA and Se (and vitamin B12), chub mackerel (*Scomber colias*) is an excellent source [11]. This fish species displays a seasonally variable DHA content that may range from high levels, such as 349.9 ± 2.5 mg/100 g ww in April, to very high levels, reaching 2473.8 mg/100 g ww in September [11]. I may be supplied by a brown seaweed, such as *Saccorhiza polyschides*, which may contain 36.7–52.2 mg/100 g dw of I [12].

For a better evaluation of nutrient dosages and food choices, it is crucial that, instead of focusing only on the total levels of target compounds from raw foods, one considers not only the effect of the culinary treatment, but also the concentration of nutrients available for intestinal absorption after digestion—bioaccessible compound content [13]). Bioaccessibility can be determined by the in vitro simulation of human digestion [13]. A recent bioaccessibility study on chub mackerel [13] has reported high Se, 91–95%, and moderately high DHA, 48–81%, bioaccessibility. Regarding I in *S. polyschides*, its bioaccessibility percentage may vary in the 30–50% range [12]. Of course, bioaccessibility also depends on culinary treatment. Culinary procedures are associated with essential components’ loss and protein denaturation, thus requiring specific studies [14]. All these aspects have to be considered for the successful development and characterization of a functional food.

Against this background, this research aims to evaluate a novel functional food in the form of a hamburger—preferred to a supplement that would require extensive processing and refining in order to supply the same amounts of neuroprotective nutrients in a small portion—as a tool for the prevention of cognitive aging, with a specific focus on the bioaccessible fatty acid profile and key nutrients (DHA, Se, and I) that have been shown to be relevant to AD prevention. To achieve this goal, the effect of different culinary treatments (steaming, roasting, and grilling) on the proximate composition, lipid fraction, and FA profile of the hamburger, as well as on its Se and I contents, while considering the bioaccessibility of these compounds, will be investigated. The resulting nutritional value and potential health benefits of the hamburger will be assessed with the goal of establishing it as a viable functional product.

## 2. Materials and Methods

### 2.1. Chub Mackerel and Seaweed

Chub mackerel (approximately 30 kg), caught off the Portuguese Coast (North Atlantic), was purchased from a local ship owner from Peniche (Portugal), in September 2021. Then, the fish were frozen and kept in isothermal boxes with ice, until being transported to the IPMA laboratory in Lisbon (Portugal). After thawing, the fish was beheaded and eviscerated. Only the edible portion (fillet, comprising both white and red muscles of the fish) was used and minced (Grindomix GM 200, Dusseldorf, Germany). Afterwards, samples were vacuum-sealed in plastic bags with a model A300/52 vacuum packager (Multivac Sepp Haggenmüller, Wolfertschwenden, Germany) and stored at −80 °C until further analysis.

The source, collection, cultivation, and preparation of the seaweed (*Saccorhiza polyschides*) is described by Cardoso et al. (2023) [12]. Briefly, the seaweed was harvested in July 2021 from an offshore Pilot Area for Aquaculture Production (APPA Armona, Olhão, Portugal). Frozen seaweed samples were transferred to the IPMA laboratory in Lisbon, where the seaweed fronds were rinsed with freshwater water to remove any remaining epiphytes and detritus. Finally, the seaweed was freeze-dried, ground, and stored individually at −80 °C until further analysis.

### 2.2. Preparation of the Functional Hamburger

The innovative functional hamburger was based on preliminary work on fish burgers, which proved the feasibility of using a high share of fish muscle in such a product without any deleterious effect on the sensory characteristics (namely, taste and texture). Given the core concept of this product, to develop a functional food for the prevention and delay of cognitive decline in the elderly population, chub mackerel—rich in DHA and Se—and seaweed—rich in I—were combined as key ingredients (Table 1). The other ingredients, purchased in a local supermarket on the day of product preparation, included quinoa, onion, garlic, red pepper, salt, thyme, oregano, parsley, and coriander, with their inclusion % in the formulation being a result of the consultation of multiple sources and preliminary testing.

For the preparation of the novel fish burgers, while the chub mackerel was used raw after mincing (Grindomix GM 200, Dusseldorf, Germany), the quinoa (*Chenopodium quinoa*) was first placed in cold water (for two hours, with the water being changed five times), washed to remove saponins and phytic acid, and then boiled (for 20 min) to reduce soluble oxalate content. The remaining ingredients were weighed and minced together. All ingredients were mixed in the previously defined proportions into a homogeneous dough that was used to mold 75 g hamburgers (around 1.5 cm high and 8.5 cm in diameter).

Four sets of hamburgers were separated and subjected to either no cooking procedure (raw) or to one of three culinary treatments: steaming, roasting, or grilling (Figure 1).

For the steaming treatment, each hamburger was wrapped in aluminum foil and cooked for 10 min with wet steam at 100 °C in a Rational CM6 oven (Rational, Landsberg, Germany). For roasting, the hamburger was also wrapped in an aluminum foil and then cooked for 10 min with dry steam at 180 °C in the same oven. Finally, the grilling process was carried out in a domestic griller (Flama, 230 V, 50 Hz, 2000 W) operated at about 180 °C for 10 min (5 min for each side of the hamburger).

After cooking, the hamburgers were homogenized and each set was separated into two sub-sets: one stored at −80 °C and the other, after being frozen at −80 °C, was freeze-dried for 48 h at −45 °C. All samples were stored under vacuum at −80 °C until further analysis.

### 2.3. Moisture, Protein, Ash, Carbohydrate, and Energy Values

The moisture and ash contents were determined according to AOAC methods [15]. The protein level was quantified according to the Dumas method [15] and a conversion factor of nitrogen into protein of 6.25 was used. This protein determination method uses equipment that enabled us to convert all N forms into gaseous nitrogen oxides (NO_x_) by complete combustion in an induction furnace, then reducing the NO_x_ gases to N_2_, and quantifying N_2_ by thermal conductivity. Carbohydrate content was estimated by difference. The energy value, expressed as kcal/100 g of edible part, were estimated: protein, 4.27 kcal/g; lipid, 9.02 kcal/g; and carbohydrate, 4.11 kcal/g [15].

### 2.4. In Vitro Digestion Model

An in vitro digestion model was chosen for the determination of FA, Se, and I bioaccessibility in the relevant ingredients—only chub mackerel in the case of FA—and in the functional hamburger subjected to different culinary treatments (raw, steamed, roasted, or grilled). Such a model comprises three sections, which enable the simulation of human digestion in three different parts of the gastrointestinal (GI) tract, the mouth, stomach, and small intestine, as described by Afonso et al. (2023) [13].

#### Calculation of Bioaccessibility

The percentage (%) of each FA in the bioaccessible fraction was estimated as follows:% FA bioaccessible = [FA]_bioaccessible_ × 100/[S](1)

Being:[FA] = Concentration of FA;[S] = [FA] before digestion.

The percentage (%) of each studied element (E) in the bioaccessible fraction was estimated as follows:% E bioaccessible = [E]_bioaccessible_ × 100/[T](2)

Being:[E] = Concentration of the element;[T] = [E] in the bioaccessible fraction + [E] in the non-digested fraction.

### 2.5. Lipid Determination

The extraction of fat from the hamburger products, boiled quinoa, and chub mackerel was performed by the Folch technique [16]. This involves weighing 100 mg of ground sample into 15 mL screw-on glass tubes with lids, adding 3 mL of chloroform-methanol solution (2:1) (*v*/*v*), and shaking lightly in the vortex for 5 min (low rotation). Afterwards, 3 mL of 0.1 N HCl and 300 µL of 0.5% MgCl_2_ were added in order to precipitate protein. After centrifugation at 2000× *g* for 5 min, the organic phase was collected with a 200 µL micropipette into another previously tared 15 mL tube and a second extraction was performed. After collecting all organic phases, they were evaporated under N stream and the lipid content was determined gravimetrically.

The procedure used for fat extraction in the bioaccessible (digested material) fraction followed Costa et al. (2015) [15]. The lipid samples were stored at −80 °C until analysis.

### 2.6. Lipid Class Determination

The relative weight of each lipid class was determined by analytical thin-layer chromatography (TLC) using a previously described method [11]. Lipid class identification was performed by comparison with standards (Sigma Chemical Co., St. Louis, MO, USA). The relative percentage of each lipid class was determined using a GS-800 densitometer and Quantity One Analysis 4.3.0 version of the software from Bio-Rad (Hercules, CA, USA).

### 2.7. Fatty Acid Profile

Fatty acid methyl esters (FAMEs) were prepared from the freeze-dried ingredients, experimental products, and bioaccessible fractions by acid-catalyzed transesterification [11]. The samples were applied to a DB-WAX (Agilent, Santa Clara, CA, USA) capillary column (film thickness: 0.25 μm, 30 m × 0.25 mm i.d.) integrated into a Varian Star 3800 CP gas chromatograph (Walnut Creek, CA, USA). The FAMEs were identified by comparing their retention times with those of several Sigma-Aldrich standards (PUFA-3, Menhaden oil, and PUFA-1, a marine source standard from Supelco, Bellefonte, PA, USA). The LOD was 1 mg/100 g. Data in mg/100 g of edible parts were calculated using the peak area ratio (% of total FA) and the lipid conversion factors [17]. Analyses were always performed in triplicate.

For an assessment of the dietary impact of the FA composition on the incidence of coronary heart disease, the index of atherogenicity (IA) and the index of thrombogenicity (IT) were calculated as proposed by Ulbricht and Southgate (1991) [18]:IA = ([12:0] + 4 × [14:0] + [16:0])/([n-6 PUFA] + [n-3 PUFA] + [MUFA]); IT = ([14:0] + [16:0] + [18:0])/(0.5 × [MUFA] + 0.5 × [n-6 PUFA] + 3 × [n-3 PUFA] + [n-3 PUFA]/[n-6 PUFA])(3)
where MUFA is the monounsaturated FA.

### 2.8. Selenium and Iodine Determination

For Se and I analyses, samples were prepared as previously described [13]. Selenium and I were measured by inductively coupled plasma mass spectrometry—ICP-MS Thermo X (Thermo Fisher, Waltham, MA, USA). The Se and I contents of samples were expressed in μg/100 g of wet weight and, in the case of the seaweed, in μg/100 g of dry weight. Analysis was performed in triplicate.

### 2.9. Polyphenolic Compound Determination

Total polyphenol content was determined by the Singleton and Rossi method using the Folin–Ciocalteu reagent [19]. Gallic acid (GA) was used as standard and phenolic content was expressed as gallic acid equivalents (mg GAE/g of dry or wet weight) through the calibration curve of gallic acid (Sigma, Steinheim, Germany). Analysis was performed in triplicate.

### 2.10. Antioxidant Activity

The Ferric Ion Reducing Antioxidant Power (FRAP) method was applied to assess the antioxidant activity, and it was based on a modified technique [20]—absorbance was read at 595 nm instead of 593 nm. Results were expressed in μmol Fe^2+^ equivalents per g of dry or wet weight and compared with an ascorbic acid control. Analysis was performed in triplicate.

### 2.11. Statistical Analysis

To test the normality and the homogeneity of variance of the data, the Kolmogorov–Smirnov test and Levene’s F-test, respectively, were used. The data were analyzed by a one-way ANOVA distribution using the Tukey HSD post hoc test to determine significant differences. For all statistical tests, the significance level (α) was 0.05. Data analysis was performed using STATISTICA 6 (Stat-sof, Inc., Tulsa, OK, USA, 2003).

## 3. Results and Discussion

Prior to the overview and discussion of the results attained with the innovative fish and seaweed burger, it is important to stress the advantages of opting for such a functional food instead of a concentrated nutritional supplement. In particular, a mackerel and seaweed hamburger may provide the neuroprotective ingredients included in ingredients naturally rich in them, thus avoiding problematic extraction/refining processes that are usually performed in the preparation of a supplement and may leave residues. In this regard, it may be noted that other similar compounds, for instance, other healthy FAs, which may have a positive synergistic interaction with other compounds are also kept in the product. Moreover, a complex food such as a burger is better suited to bring together chemically diverse substances (for instance, in a supplement, lipophilic and hydrophilic molecules would need an emulsifying agent to be kept stably together). Finally, there is the reason of consumer perception, since an elderly population will more easily accept and consume traditional food on a regular basis.

### 3.1. Proximate Composition

The results show that, while chub mackerel is rich in lipid (7.22 ± 1.32%, ww) and protein (22.7 ± 0.1%, ww) contents, quinoa contains a high level of carbohydrates, exceeding 16%, ww (Table 2). Moreover, this plant ingredient is very low in fat (0.27 ± 0.05%, ww) and protein (2.70 ± 0.21%, ww) contents. The hamburgers had a nutritional profile more similar to chub mackerel, but with a lower protein ≤ 22%, ww content. The culinary treatment affected the proximate composition. With respect to the raw product, the moisture content declined in the cases of roasted and grilled hamburgers, from 71.5 ± 0.3%, ww to 67.8 ± 0.4%, ww and 64.0 ± 0.2%, ww, respectively. On the other hand, the roasted and grilled functional food was richer in protein than the raw product, 20.0 ± 0.3%, ww and 22.0 ± 0.2%, ww, respectively, vs. 18.0 ± 0.6%, ww (raw hamburger). The same relative enrichment with roasting and grilling was observed in the case of the ash content. Only the lipid content remained unaltered with the different cooking procedures. Finally, concerning the energy value, hamburgers were in a 133–169 kcal/100 g range, thus proving to be generally as caloric as raw chub mackerel, but more caloric than quinoa.

The hamburgers’ proximate composition reflected the nutritional profile of chub mackerel and quinoa, which are quantitatively the two most important ingredients in the functional food (a total sum exceeding 88%, ww). This was particularly clear in the raw product, since its moisture, lipid, protein, and carbohydrate percentages were proportional to the fish (moisture, lipid, and protein) and quinoa (moisture and carbohydrate) contributions. The relatively high ash content in the burgers must be ascribed to the salt and other ingredients. Moreover, the culinary treatments influenced the moisture, protein, and ash contents. Grilling was clearly the most drastic thermal treatment, leading to the greatest moisture loss and, concomitantly, to the highest protein and ash gains in percentage. Of course, the calculated energy value followed this trend. It should be stressed that the cooked hamburgers had an energy content similar to chub mackerel and in a range (140–170 kcal/100 g) of low-calorie density foods, thereby contrasting with the growing number of high-calorie-density foods in the Western diet [21].

### 3.2. Lipid Classes

Regarding the ingredients, while chub mackerel was relatively rich in triacylglycerols (TAGs), quinoa had a more mixed and complex set of lipid classes (Table 3). The share of free fatty acids (FFAs) was quite high in the quinoa, reaching 23.9 ± 2.7% of the total lipids. In the products, the TAG value was higher, varying between 63.6 ± 2.5% in the roasted hamburger and 67.4 ± 0.9% in the grilled hamburger. Conversely, FFAs were lower in the grilled product, 8.9 ± 0.7%, than in the roasted one, 12.7 ± 1.7%. Concerning phospholipids (PLs), the steamed burger had a larger share of these lipids, 10.2 ± 0.8%, than both the roasted and grilled burgers, 8.0–8.4%. None of the products’ PLs differed from those determined in the chub mackerel.

After digestion, TAG hydrolysis was complete in both chub mackerel and the various hamburger products. This led to the formation of FFAs, which constituted the main lipid class in all bioaccessible fractions, in the range of 45.7 ± 1.2% in the roasted hamburger to 51.5–51.7% in the steamed hamburger and raw mackerel. The diacylglycerols (DAGs) that were present in the initial samples were partially hydrolyzed. Monoacylglycerols (MAGs) were formed as a result of TAG and DAG hydrolysis, reaching a higher share in the fish burgers, 28.0–30.1%, than in the chub mackerel, 25.0 ± 0.7%.

It must be mentioned that there was some hydrolytic degradation in the chub mackerel, since, usually, the TAG share of the total lipids exceeds 80–90% in fatty fish [15], whose storage lipids (TAGs) tend to be much more important and fluctuate more intensely with the total lipid content than the structural lipids (PLs) [22]. The hamburgers seemed to be less affected by this hydrolysis, but their TAG levels were lower than would be expected if the raw material was in perfect condition—this is what often happens in real industrial conditions. TAG hydrolysis during the in vitro digestion led to the disappearance of the TAG band and to the formation of FFAs and MAGs. This is fully within what would be expected, based on previous work [15], and signals complete and successful lipid digestion.

### 3.3. Fatty Acid Profile

Regarding the relative values, whereas quinoa differed from all other samples, there were relevant similarities between the hamburgers and chub mackerel (Table 4). In fact, if major FA classes are considered, quinoa was poorer in saturated FA (SFA) and n-3 PUFA and richer in MUFA and n-6 PUFA than the other samples. It can be highlighted that, while the n-3 PUFA content was only 6.0 ± 0.4%, the n-6 PUFA content reached more than 50%. This contrasts starkly with the chub mackerel and hamburgers, which had high n-3 PUFA contents in the 42.0–46.4% range and low n-6 PUFA contents in the range of 6.6–6.9%. As a result, the n-3/n-6 ratio was extremely low in quinoa (<1) and quite high in the mackerel and the fish burgers (>6). Such differences were also reflected by the IA parameter, which was lower in the quinoa (not exceeding 0.1), but otherwise also low in the other samples: 0.2–0.3.

The differences observed between the FA profile of quinoa and those of the other samples were as follows: stearic (18:0), oleic (18:1n-9), linoleic (18:2n-6), eicosapentaenoic (EPA, 20:5n-3), and docosapentaenoic (DPA, 22:5n-3) acids, as well as DHA. Indeed, while the quinoa was much richer in oleic and linoleic acids than the other samples, 20.8 ± 0.7% vs. 12.3–13.7% and 52.6 ± 3.4% vs. 2.2–2.7%, this pseudocereal had a lower content of long-chain SFA stearic acid, 1.4 ± 0.5% vs. 6.2–6.5%, and did not present any quantifiable levels of EPA, DPA, and DHA.

With regard to the other samples, two major aspects are worth analyzing: (i) the dichotomy between the main ingredient (>70% of the hamburger weight), chub mackerel, and the fish burgers; and (ii) the effect of the culinary treatment on the FA composition. Concerning the former aspect, no large difference was observed for any of the major FA groups (SFA, MUFA, PUFA, n-3 PUFA, or n-6 PUFA). Accordingly, no difference was detected between the n-3/n-6 ratio of the chub mackerel and the ratios of the various hamburgers. After a more painstaking analysis of the results, it was remarkable to see that no significant difference was detected in the various FA contents, with the exception of slight variations in the levels of rather secondary FAs, such as 20:4n-3, 21:5n-6, and 22:4n-6. Regarding (ii), among the major groups and n-3/n-6 ratio, there was no difference, and for IA and IT, the values were all low, with only slight variations. Namely, the steamed hamburger had slightly higher IA and IT values than the roasted and grilled products. Likewise, at a more detailed level, only one FA showed significant differences in its relative importance in different hamburger samples: 21:5n-3, the content of which in raw and steamed hamburgers was lower than in the roasted hamburger: 0.3 ± 0.0% vs. 0.4 ± 0.1%.

The absolute contents displayed a higher level of variability between samples (Table 4). In the case of quinoa, as a result of its very low lipid content, 0.27 ± 0.05% (Table 2), all FAs exhibited absolute concentrations lower than the other samples with the sole exception of linoleic acid. For the other major ingredient, chub mackerel, almost all FA concentrations exceeded those determined in the hamburgers. This was also observed in the major groupings: SFA, MUFA, PUFA, n-3 PUFA, and n-6 PUFA. Only in the cases of myristic acid (14:0), linoleic acid, and α-linolenic acid (18:3n-3), no difference was registered. With respect to cooking, these three FA contents were also unchanged. However, the absolute concentration of most FAs (in mg/100 g ww) was affected by culinary treatment. Namely, the total MUFA level increased after cooking, regardless of the specific treatment: 1080–1163 mg/100 g ww vs. 917 ± 53 mg/100 g ww. The same effect was also observed in the case of two relevant FAs: stearic and oleic acids. For other FAs and major groups, roasting and grilling influenced their levels, but not steaming. This was observed for total PUFA, 2777–2805 mg/100 g ww in the roasted and grilled samples vs. 2229 ± 105 mg/100 g ww in the raw sample, and n-3 PUFA, 2374–2489 mg/100 g ww in the roasted and grilled samples vs. 1909 ± 99 mg/100 g ww in the raw sample. This concentration increase after roasting and grilling was also measured for 17:0, 20:1n-9, 20:2n-6, arachidonic acid (20:4n-6), EPA, 21:5 n-3, 22:5 n-6, DPA, and DHA. In the latter case, while the raw hamburger contained 1227 ± 92 mg/100 g ww, the roasted and grilled products were 1544–1552 mg/100 g ww. Furthermore, in some instances, only the grilled hamburger had a higher content than the raw product. An isolated effect of grilling on FA levels was observed in the following cases: SFA,1208 ± 111 mg/100 g ww vs. 950 ± 45 mg/100 g ww, n-6 PUFA, 365 ± 4 mg/100 g ww vs. 291 ± 7 mg/100 g ww, 17:1, 25 ± 1 mg/100 g ww vs. 21 ± 1 mg/100 g ww, stearidonic acid (18:4n-3), 63 ± 4 mg/100 g ww vs. 51 ± 3 mg/100 g ww, and 20:4 n-3, 32 ± 2 mg/100 g ww vs. 26 ± 1 mg/100 g ww. No such isolated effect was detected in the case of roasting and only 16:1n-9 provided an instance of an isolated effect of steaming on the FA content.

The bioaccessible FA profiles of chub mackerel and the functional foods also showed significant differences (Table 5). In the case of SFA and MUFA, determined contents in fish were generally higher than in the raw hamburger, in particular, for stearic acid (258 ± 13 mg/100 g ww vs. 197 ± 4 mg/100 g ww), oleic acid (646 ± 24 mg/100 g ww vs. 530 ± 36 mg/100 g ww), and the odd chain FA, 17:0 (43 ± 4 mg/100 g ww vs. 34 ± 1 mg/100 g ww) and 17:1 (24 ± 2 mg/100 g ww vs. 21 ± 1 mg/100 g ww), which reduce the risk of metabolic diseases [23]. Regarding the steamed and roasted fish burgers, a difference with higher levels in chub mackerel was only observed for stearic acid and 18:1n-7. The SFA and MUFA contents in the grilled hamburger were never lower than in the fish. In the case of PUFAs, regardless of being n-6 or n-3 FAs, the main opposition was between grilled burgers and the chub mackerel. In fact, contents were mostly higher in the grilled hamburger than in its main ingredient. Namely, the grilled product had an α-linolenic acid bioaccessible content of 55 ± 3 mg/100 g ww, which was higher than its counterpart in chub mackerel: 43 ± 10 mg/100 g ww. The grilled burger also contained 55 ± 3 mg/100 g ww of bioaccessible arachidonic acid, compared to 47 ± 2 mg/100 g ww in fish, and 441 ± 17 mg/100 g ww of bioaccessible EPA, which exceeded 323 ± 20 mg/100 g ww of bioaccessible EPA in fish. The same trend was observed in other PUFAs, including DPA and DHA.

Among the fish burgers, most differences in the bioaccessible fractions were observed between the raw and grilled products. Indeed, this was the case for all studied FAs with the exception of 17:0, α-linolenic acid, and 22:4n-6. For all the other FAs, with exception of linoleic acid, their bioaccessible content was higher in the grilled hamburger than the uncooked one. For instance, bioaccessible DHA in the grilled hamburger was 1219 ± 95 mg/100 g ww, thus clearly above the same content in the raw hamburger: 938 ± 66 mg/100 g ww. A higher bioaccessible content in the raw product was only measured in the case of linoleic acid: 16 ± 0 mg/100 g ww vs. 8–11 mg/100 g ww (range in all other burgers). In some cases, steaming and roasting also led to a bioaccessible concentration increase with respect to the raw product: palmitic acid, stearic acid, 17:1, 18:1 (n-9 and n-7), stearidonic acid, and EPA in the steamed burger and 18:1 (n-9 and n-7) in the roasted burger.

The calculated bioaccessible percentage varied widely across type of FA and sample. All FAs present in fish displayed a bioaccessibility that was lower than at least one of the fish burgers, which reflected the lower lipid bioaccessibility in the chub mackerel: 73 ± 8% vs. 84–96%. Some examples may be provided, such as oleic acid, 76 ± 2% (chub mackerel) vs. >90% (burgers); stearidonic acid, 57 ± 4% vs. 95–100%; arachidonic acid, 30 ± 1% vs. 36–48%; EPA, 47 ± 3% vs. 74–86%; and DHA, 41 ± 3% vs. 62–79%. Bioaccessibility was also affected by culinary treatment, but with fewer differences than between the fish and burgers. Namely, stearic acid, α-linolenic acid, arachidonic acid, and EPA in the grilled hamburger were more bioaccessible than in the roasted hamburger. This difference was mostly small, such as in EPA, 82 ± 3% vs. 74 ± 1%, or arachidonic acid, 44 ± 2% vs. 36 ± 1%. Only in the case of stearic acid was a large difference of 27% between the grilled and roasted products calculated.

The FA profiles of chub mackerel and quinoa correspond to what has been reported in the literature [17,24]. This agreement comprises the high EPA and DHA contents in chub mackerel and the >50% linoleic acid content in quinoa. Despite this very high share of linoleic acid and the high incorporation rate of quinoa in the burgers, approximately 16% (Table 1), the linoleic acid level in the hamburgers did not differ from the relatively low content in fish. This is related to the very low lipid content in quinoa, which is also noted when absolute FA contents are verified. This means that quinoa lipids made a small contribution to the FA profile of the fish burgers, with this mainly being determined by the chub mackerel’s contribution. This was corroborated by practically all FA contents. The absolute levels in the hamburgers were usually lower than those in the chub mackerel due to the diluting effect of the other ingredients, which were, on average, poorer in lipids.

The relative FA levels did not show a strong effect of the culinary treatment on them. This means that cooking, even in its more thermally drastic form (as in grilling), did not degrade the most thermo-sensitive FAs, PUFAs, especially those with a higher number of double bonds that can undergo isomerization reactions and other chemical alterations [25]. The effect of temperature on the FA composition is a matter of contention, with different findings by distinct research groups [15,26,27,28]. This divergence may derive from different temperatures and times for a given culinary treatment and species-specific biochemical aspects. In fact, there are studies supporting a negative effect, especially of grilling, on EPA, DHA, and other PUFAs [26,27]. For instance, Choo et al. (2018) [24] concluded that deep frying and grilling methods showed a significant reduction in DHA and EPA contents in all fish fillets compared with steaming and baking in foil. In addition, Schneedorferová et al. (2015) [27] found a drastic decrease in the n-3/n-6 ratio in marine fish after grilling, with oven-baking being the mildest heat treatment for PUFA preservation. These authors also reported impact variability across species, with herring being the most heat-stable from the tested species. On the other hand, other studies did not observe any impacts, such as in a study on silver catfish [28], which concluded that boiling, baking, and grilling did not affect the fillets’ FA composition. Costa et al. (2015) [15] reported that the FA proportions determined for grilled, steamed, and roasted salmon were similar to those observed in raw salmon. These authors also observed that differences in absolute values were due to the lipid content variation with culinary treatment, resulting in the highest contents in grilled salmon. Considering also moisture loss, this agrees with the current study on fish hamburgers. Regarding this subject, it should also be noted that the hamburgers contained other ingredients besides fish, which could provide protection from oxidation to the most sensitive FA. Indeed, Wang et al. (2018) [29] found that rosemary—a plant from the same family of thyme and oregano, used as ingredients (Table 1)—extract had a positive effect on the stability of flaxseed oil against oxidation, thereby paving the way for future applications for the protection of n-3 FAs in food. Moreover, Serdaroğlu and Felekoğlu (2005) [30] observed a positive effect of using rosemary extract and onion juice—with onion being an important hamburger ingredient—on the oxidative stability of sardine mince.

The differences between the bioaccessible FA concentrations in fish and hamburgers resulted from the variability in the initial (prior to digestion) absolute concentrations and the specific bioaccessibility of each FA, which, in turn, varied between chub mackerel and hamburgers as well as between raw and cooked products. However, this latter variability was limited even between raw and grilled hamburgers. In some instances, the bioaccessibility percentages of some FAs were higher after grilling (e.g., stearic acid), which diverges from other experimental work on cooked fish [15]. For instance, Costa et al. (2015) [15] reported the lower bioaccessibility of stearic acid after grilling salmon, which may be related to lower protein bioaccessibility in grilled fish due to protein denaturation and aggregation triggered by high temperatures. However, hamburgers with their complex mixture of ingredients may present other phenomena that warrant further research.

In general, the bioaccessibility values of main SFAs and MUFAs were higher than those of key PUFAs, such as EPA and DHA. This agrees with previous findings [15]. In any case, with the exception of linoleic acid, all major FAs had moderately high to very high bioaccessibility percentages in the fish burgers. Lower bioaccessibility levels were observed only in chub mackerel, thus entailing bioaccessibility enhancement with the mincing and ingredient mixing that was performed for the hamburgers’ preparation. Such preparation may release fish lipids from their original structures and cause them to make contact with other components—which may include emulsifying agents—from all the ingredients. This interpretation may be partially corroborated by Wang et al. (2022) [31], who found higher DHA bioaccessibility when its oil was encapsulated with whey proteins as Pickering emulsion, and by Lin et al. (2014) [32], who inferred higher DHA bioaccessibility as a result of oil emulsification with soy lecithin.

From a nutritional point of view, absolute FA levels before and after digestion simulations may be used to investigate whether these hamburgers convey benefits to human health. Taking into account human EPA + DHA requirements, the amount of cooked—and raw, as a comparative reference—fish burger required to meet the adequate iIntake (AI) of EPA + DHA (250 mg/day) according to the EFSA Panel on Dietetic Products, Nutrition and Allergies and its established Dietary Reference Values (DRVs) for the EU population may be calculated [33]. Assuming initial contents, 15.3 g raw, 15.0 g steamed, 12.3 g roasted, and 12.1 g grilled hamburger quantities would meet the DRVs. In the latter two cases, an 80–85 g single meal every week would ensure EPA + DHA requirements. However, since EPA and DHA are not totally available for intestinal absorption, calculations can be performed with bioaccessible contents, thus yielding 19.6 g raw, 18.6 g steamed, 18.9 g roasted, and 15.1 g grilled hamburger quantities for meeting the RDI. These are perfectly viable amounts. For instance, a single 150 g weekly meal of this functional food, regardless of being steamed, roasted, or grilled, would provide the EPA and DHA needs. Hence, this functional hamburger would be a suitable source of EPA and DHA in the human diet. DPA as precursor of many major lipid mediators (protectins, resolvins, maresins, and isoprostanoids) involved in the pro-resolution of inflammation, with specific effects compared to other n-3 PUFAs, may also be highlighted as a positive health benefit, given DPA’s substantial levels and high bioaccessibility in the hamburgers. The potential benefit to human health is corroborated by the low IA and IT values, which are close to the values stated for the so-called Eskimo diet, pointing to a very low incidence of coronary heart disease (IA: 0.39; IT: 0.28) [18]. Moreover, odd chain FAs, 17:0 and 17:1, even if present at low concentrations, are important as potential contributors to the reduction in the risk of metabolic diseases [23].

### 3.4. Selenium and Iodine Contents

Compared to quinoa and seaweed, the chub mackerel proved to be a good source of Se with a total content of 59 ± 4 μg/100 g ww, but not of I, with only 25 ± 1 μg/100 g ww (Table 6). However, the freeze-dried seaweed (*S. polyschides*), another hamburger ingredient, was very rich in I: 36,720 ± 603 μg/100 g dw. This brown seaweed also contained Se > 100 g/100 g dw. Quinoa was not a good source of any of these elements. Regarding the hamburgers themselves, the initial Se and I levels showed some differences as a result of specific culinary treatments. Both elements exhibited a contrast between the raw and steamed hamburgers and the grilled hamburger, with the roasted product being in an intermediate position. Indeed, grilled fish burgers contained more Se and I than the raw and steamed ones (61 ± 5 μg/100 g ww vs. 48–50 μg/100 g ww and 255 ± 6 μg/100 g ww vs. 221–224 μg/100 g ww, respectively).

The bioaccessible Se and I concentrations were always significantly lower than the initial ones, resulting in bioaccessibility percentages below 90%. In the case of Se, no difference was identified as a result of the different cooking procedures. For I, the raw burgers had less bioaccessible elements than all the other products: 127 ± 4 μg/100 g ww vs. 153–163 μg/100 g ww. Bioaccessibility percentages varied between products as well as with respect to the ingredients. Namely, Se bioaccessibility in quinoa was distinctly lower than in raw mackerel and the hamburgers, <10% vs. 70–85%, and I bioaccessibility in the seaweed was the lowest of all samples: 48 ± 3% vs. 57–78%. On the other hand, the highest I bioaccessibility percentage was calculated for the raw chub mackerel: 78 ± 0%. The fish burgers can be divided in two groups, raw and grilled products, with 57–60% I bioaccessibility and steamed and roasted products with 69–70% I bioaccessibility.

The high value of Se in the chub mackerel is within the previously reported range of concentrations [13]. The same applies to I in *S. polyschides* [12]. The Se and I contents in the raw hamburger reflect the proportions of fish and seaweed in the product formulation (Table 1). The concentration increment with grilling in comparison to the raw hamburger can be ascribed to the observed moisture loss due to this more drastic thermal treatment (Table 2).

The high Se bioaccessibility in chub mackerel and in fish burgers agrees with the previous findings [13]. It is possible that Se is mainly bound to fish proteins, which are readily available to the digestive enzymes used in the in vitro methodology [13]. The relatively low I bioaccessibility in the seaweed is similar to that calculated by Cardoso et al. (2023) [12], being a possible consequence of human digestive enzymes’ inability to hydrolyze many of the polysaccharides that exist in seaweed biomass [34]. In addition, the relatively high I bioaccessibility in fish and derived products agrees partly with Alves et al. (2018) [35], who determined I bioaccessibility levels of approximately 80% in mussel and 50% in tuna. Ferraris et al. (2021) [36] calculated I bioaccessibility levels of almost 100% in blue whiting. This elemental bioaccessibility has been related to I species present in the sample, being organically bound I less bioaccessible than iodide [36,37]. However, since seaweed was the major contributor of I to the fish burgers, the higher bioaccessibility in the products with respect to *S. polyschides* seems to indicate a release of bound I after mixing the freeze-dried seaweed with the other ingredients.

With Se and I being essential elements, it is possible to calculate how much fish burger would be needed to meet the Se and I adequate intakes (Ais) [33]. Whereas Se AI for >18-year-old adults (with the exception of pregnant and lactating women) is set at 70 μg/day, the advised I AI is 150 μg/day [33]. For Se, 145.9 g raw, 140.0 g steamed, 129.7 g roasted, and 114.8 g grilled hamburger quantities would meet the Se AI values. Considering bioaccessibility, 184.2 g raw, 170.7 g steamed, 152.2 g roasted, and 162.8 g grilled hamburger quantities would be necessary. Of course, a daily consumption of 150–180 g of fish burger would not occur in reality, but these results indicate that these hamburgers are a good source of Se. As for I, 67.0 g raw, 67.9 g steamed, 64.7 g roasted, and 58.8 g grilled hamburger quantities would ensure the I DRI. After the bioaccessibility input, these values can be recalculated: 118.1 g raw, 98.0 g steamed, 92.0 g roasted, and 98.0 g grilled hamburgers. Hence, the novel functional burger is also a rich source of I.

### 3.5. Polyphenol Content and Antioxidant Activity

Both the polyphenol content and antioxidant activity (measured by the FRAP method) of seaweed and fish burgers are displayed in Table 7. The functional product had lower phenolic contents and antioxidant activities (in wet weight) than the freeze-dried seaweed. However, even taking into account the moisture levels in the fish burgers (Table 2), these were clearly less rich in poylphenols and antioxidant activity than the brown seaweed used as their ingredient. Though there were other vegetable ingredients known to be rich in polyphenols and antioxidants in their formulation, such as oregano (Table 1) [38], chub mackerel, which is not considered a source of polyphenols [39], had a diluting effect and, as such, may have led to the lower antioxidant activity in the fish burgers. Only a much higher incorporation rate of *S. polyschides* in the products could have imparted relevant antioxidant properties to them. For this reason, the phenolic content of the produced mackerel burgers, 0.02–0.12 mg GAE/g ww, is only comparable to tuna burgers without any specific enrichment: 0.47–0.51 mg GAE/g dw [40]. These same burgers after enrichment with dry olive paste flour reached 6.15–6.55 mg GAE/g dw. However, incorporation rates apparently reached 10% ww in the tuna burgers [40]. For fish burgers with 1% dried *Cystoseira compressa* (a brown seaweed), there was also a modest increase in FRAP with respect to the control burger [41]. Finally, it may be noted that grilling had a deleterious effect on the polyphenol content and antioxidant activity of the mackerel burgers. This was also observed in the case of grilled vs. raw beef hamburgers enriched with a phenolic extract: 0.020–0.049 vs. 0.085–0.168 mg phenolic compounds/g ww [42].

## 4. Conclusions

The developed hamburgers showed important composition differences. The roasted and grilled fish burgers had a lower moisture level and higher protein content, while the FA profile varied in absolute values, but not in relative values, between products. From a nutritional standpoint, the hamburgers had high n-3 PUFA contents and low n-6 PUFA contents, yielding high n-3/n-6 ratios, and could meet human EPA + DHA requirements. The bioaccessibility of FAs varied by type and sample. Polyphenol enrichment by the seaweed and antioxidant activity were limited in all burgers. The initial Se and I levels in the functional food were high, and Se bioaccessibility was high in the hamburgers, with I bioaccessibility somewhat lower. This functional hamburger could be a suitable source of EPA, DHA, Se, and I in the human diet, thereby potentially providing biologically active and bioaccessible nutrients for the prevention of cognitive aging. Given the moisture-dependent nutrient contents and notwithstanding the low phenolic content, the grilled hamburger would offer the highest amount of beneficial nutrients per 100 g, with 162.8 g of this food being enough to meet the dietary requirements of Se, I, and EPA + DHA. Moreover, based on the low IA and IT values (comparable to reference values associated with a very low incidence of coronary heart disease), this functional food may also contribute to the prevention of cardiovascular disease.

## Figures and Tables

**Figure 1 foods-13-01332-f001:**
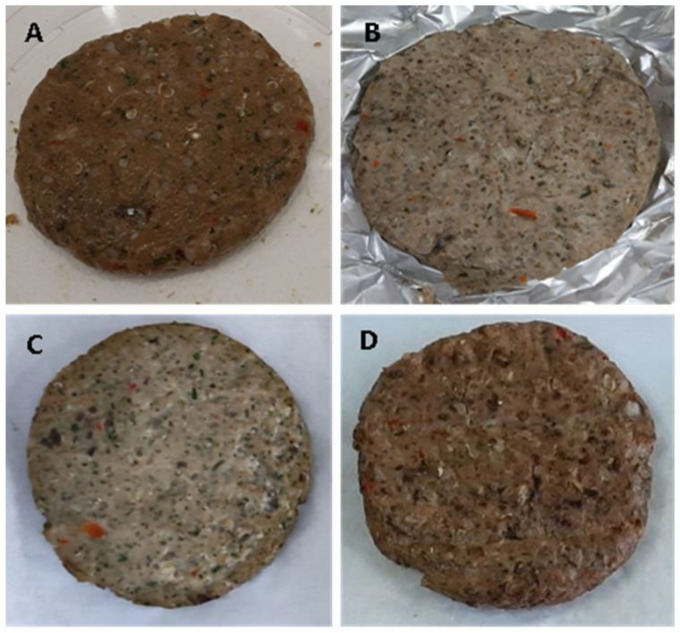
The prepared fish hamburgers: (**A**) raw; (**B**) steamed; (**C**) roasted; and (**D**) grilled.

**Table 1 foods-13-01332-t001:** Ingredients (%, ww) used in the formulation of the chub mackerel hamburger.

Ingredient	(%, ww)
Chub mackerel	72.43
Boiled quinoa	15.73
Onion	4.73
Red pepper	3.73
Garlic	1.60
Seaweed	0.53
Salt	0.47
Thyme	0.33
Oregano	0.16
Parsley	0.16
Coriander	0.13

**Table 2 foods-13-01332-t002:** Proximate compositions (expressed in g/100 g) of chub mackerel, cooked quinoa, and hamburgers (raw and cooked).

	Moisture (%)	Lipid (%)	Protein (%)	Ash (%)	Carbohydrates (%) *	Energy Value (kcal/100 g)
Chub mackerel (raw)	68.5 ± 0.1 ^c^	7.22 ± 1.32 ^b^	22.7 ± 0.1 ^a^	1.31 ± 0.03 ^b^	0.20	157
Quinoa (boiled)	80.2 ± 0.8 ^a^	0.27 ± 0.05 ^a^	2.70 ± 0.21 ^d^	0.20 ± 0.01 ^a^	16.63	80
Raw hamburger	71.5 ± 0.3 ^b^	5.13 ± 0.60 ^b^	18.0 ± 0.6 ^c^	1.74 ± 0.05 ^c^	3.58	133
Steamed hamburger	70.3 ± 1.0 ^bc^	5.65 ± 0.23 ^b^	18.3 ± 0.8 ^c^	1.84 ± 0.02 ^c^	3.91	140
Roasted hamburger	67.8 ± 0.4 ^c^	6.49 ± 0.30 ^b^	20.0 ± 0.3 ^b^	2.02 ± 0.02 ^d^	3.73	153
Grilled hamburger	64.0 ± 0.2 ^d^	6.71 ± 0.32 ^b^	22.0 ± 0.2 ^a^	2.21 ± 0.02 ^e^	5.06	169

Values are presented as the mean ± standard deviation. In the same column, different lowercase letters between samples represent significantly different arithmetic means (*p* < 0.05). * Calculated by difference.

**Table 3 foods-13-01332-t003:** Lipid class distributions (in % of the total lipids) of chub mackerel, boiled quinoa, and hamburgers (raw and cooked) before (Initial) and after digestion (Bioacc.).

Lipid Class (% Total Lipids)	Quinoa (Boiled)	Raw Chub Mackerel		Raw Hamburger		Steamed Hamburger		Roasted Hamburger		Grilled Hamburger	
	Initial	Initial	Bioacc.	Initial	Bioacc.	Initial	Bioacc.	Initial	Bioacc.	Initial	Bioacc.
PL	5.3 ± 0.7	9.2 ± 1.0 ^ab^	13.7 ± 0.5 ^AB^	8.9 ± 1.1 ^ab^	13.2 ± 0.7 ^AB^	10.2 ± 0.8 ^a^	12.4 ± 0.6 ^A^	8.4 ± 0.4 ^b^	16.1 ± 1.2 ^B^	8.0 ± 0.8 ^b^	14.2 ± 1.6 ^AB^
MAG	---	---	25.0 ± 0.7 ^A^	---	30.1 ± 0.3 ^B^	---	28.0 ± 1.0 ^B^	---	28.8 ± 1.2 ^B^	---	28.7 ± 0.4 ^B^
ST + 1,2-DAG	4.2 ± 0.5	18.3 ± 0.6 ^a^	9.6 ± 0.0 ^A^	12.6 ± 1.2 ^b^	8.4 ± 0.9 ^A^	11.4 ± 0.8 ^b^	8.1 ± 0.7 ^A^	15.3 ± 1.8 ^a^	9.3 ± 1.0 ^A^	15.8 ± 2.1 ^a^	8.4 ± 0.5 ^A^
FFA	23.9 ± 2.7	18.9 ± 1.0 ^a^	51.7 ± 0.8 ^A^	12.6 ± 1.9 ^b^	48.3 ± 0.6 ^AB^	11.2 ± 1.0 ^bc^	51.5 ± 2.1 ^A^	12.7 ± 1.7 ^b^	45.7 ± 1.2 ^B^	8.9 ± 0.7 ^c^	48.7 ± 0.3 ^AB^
TAG	43.9 ± 3.0	53.6 ± 2.1 ^a^	---	65.9 ± 1.7 ^bc^	---	67.2 ± 1.0 ^c^	---	63.6 ± 2.5 ^b^	---	67.4 ± 0.9 ^c^	---
Others	22.7 ± 1.7	---	---	---	---	---	---	---	---	---	---

Values are presented as mean ± standard deviation. Different lowercase letters in a row correspond to significant differences (*p* < 0.05) between the different culinary treatments (and chub mackerel). Different uppercase letters in a row correspond to significant differences (*p* < 0.05) between the different culinary treatments (and chub mackerel) in the bioaccessible fraction. PLs—phospholipids; MAGs—monoacylglycerols; ST + 1,2-DAG—sterol together with 1,2-Diacylglycerols; FFAs—free fatty acids; and TAGs—triacylglycerols.

**Table 4 foods-13-01332-t004:** Fatty acid profiles (in % of total FA and in mg/100 g ww) of chub mackerel, boiled quinoa, and hamburgers (raw and cooked).

Fatty Acid	Raw Chub Mackerel		Quinoa (Boiled)		Raw Hamburger		Steamed Hamburger		Roasted Hamburger		Grilled Hamburger	
	(%)	(mg/100 g)	(%)	(mg/100 g)	(%)	(mg/100 g)	(%)	(mg/100 g)	(%)	(mg/100 g)	(%)	(mg/100 g)
14:0	1.4 ± 0.3 ^a^	95 ± 16 ^A^	---	---	1.4 ± 0.2 ^a^	58 ± 10 ^A^	1.7 ± 0.1 ^a^	82 ± 5 ^A^	1.3 ± 0.1 ^a^	66 ± 6 ^A^	1.5 ± 0.5 ^a^	81 ± 27 ^A^
15:0	0.5 ± 0.1 ^a^	33 ± 5 ^A^	---	---	0.5 ± 0.1 ^a^	21 ± 3 ^B^	0.6 ± 0.0 ^a^	28 ± 1 ^AB^	0.5 ± 0.0 ^a^	25 ± 2 ^AB^	0.5 ± 0.1 ^a^	28 ± 6 ^AB^
16:0	11.9 ± 1.2 ^a^	784 ± 79 ^A^	8.5 ± 0.4 ^b^	21 ± 1 ^C^	12.2 ± 0.9 ^a^	510 ± 38 ^B^	13.6 ± 0.3 ^a^	635 ± 16 ^B^	11.6 ± 0.5 ^ab^	598 ± 28 ^B^	12.1 ± 1.4 ^a^	643 ± 77 ^AB^
17:0	0.7 ± 0.1 ^a^	47 ± 4 ^A^	---	---	0.8 ± 0.0 ^a^	32 ± 1 ^C^	0.8 ± 0.0 ^a^	37 ± 1 ^BC^	0.8 ± 0.0 ^a^	39 ± 1 ^B^	0.7 ± 0.0 ^a^	39 ± 2 ^B^
18:0	6.2 ± 0.3 ^a^	407 ± 20 ^A^	1.4 ± 0.5 ^b^	3.5 ± 1.2 ^E^	6.4 ± 0.2 ^a^	267 ± 6 ^D^	6.5 ± 0.1 ^a^	301 ± 4 ^C^	6.4 ± 0.0 ^a^	328 ± 1 ^BC^	6.3 ± 0.1 ^a^	335 ± 5 ^B^
ΣSFA	22.0 ± 2.0 ^a^	1453 ± 134 ^A^	11.1 ± 0.7 ^b^	27 ± 2 ^D^	22.7 ± 1.1 ^a^	950 ± 45 ^C^	24.8 ± 0.5 ^a^	1157 ± 24 ^BC^	22.0 ± 0.6 ^a^	1135 ± 33 ^BC^	22.8 ± 2.1 ^a^	1208 ± 111 ^B^
16:1n-9	0.9 ± 0.1 ^a^	61 ± 6 ^A^	---	---	0.9 ± 0.1 ^a^	37 ± 4 ^C^	1.1 ± 0.1 ^a^	49 ± 2 ^B^	0.9 ± 0.0 ^a^	48 ± 1 ^BC^	0.9 ± 0.1 ^a^	47 ± 6 ^BC^
16:1n-7	1.6 ± 0.1 ^a^	108 ± 8 ^A^	---	---	1.6 ± 0.1 ^a^	67 ± 6 ^B^	1.8 ± 0.0 ^a^	84 ± 2 ^B^	1.5 ± 0.1 ^a^	75 ± 7 ^B^	1.6 ± 0.3 ^a^	86 ± 14 ^B^
17:1	0.5 ± 0.0 ^a^	31 ± 3 ^A^	0.5 ± 0.0 ^a^	1.3 ± 0.1 ^D^	0.5 ± 0.0 ^a^	21 ± 1 ^C^	0.5 ± 0.0 ^a^	23 ± 0 ^BC^	0.5 ± 0.0 ^a^	24 ± 1 ^BC^	0.5 ± 0.0 ^a^	25 ± 1 ^B^
18:1n-9	12.3 ± 0.5 ^a^	813 ± 30 ^A^	20.8 ± 0.7 ^b^	51 ± 2 ^D^	13.0 ± 0.8 ^a^	543 ± 35 ^C^	13.7 ± 0.0 ^a^	639 ± 1 ^B^	13.2 ± 0.0 ^a^	678 ± 1 ^B^	13.0 ± 0.3 ^a^	687 ± 14 ^B^
18:1n-7	3.1 ± 0.1 ^a^	204 ± 6 ^A^	1.1 ± 0.1 ^b^	2.7 ± 0.2 ^E^	3.1 ± 0.1 ^a^	129 ± 4 ^D^	3.2 ± 0.0 ^a^	147 ± 1 ^C^	3.1 ± 0.0 ^a^	157 ± 1 ^BC^	3.0 ± 0.0 ^a^	161 ± 1 ^B^
20:1n-9	1.6 ± 0.1 ^a^	106 ± 4 ^A^	1.9 ± 0.1 ^b^	4.6 ± 0.3 ^D^	1.6 ± 0.1 ^a^	67 ± 4 ^C^	1.7 ± 0.0 ^ab^	77 ± 1 ^BC^	1.7 ± 0.2 ^ab^	88 ± 8 ^B^	1.7 ± 0.1 ^ab^	88 ± 4 ^B^
ΣMUFA	21.2 ± 0.7 ^a^	1401 ± 46 ^A^	26.0 ± 0.5 ^b^	64 ± 1 ^D^	21.9 ± 1.3 ^a^	917 ± 53 ^C^	23.1 ± 0.0 ^a^	1080 ± 2 ^B^	22.1 ± 0.2 ^a^	1139 ± 8 ^B^	21.9 ± 0.6 ^a^	1163 ± 30 ^B^
18:2n-6	2.2 ± 0.8 ^a^	147 ± 51 ^A^	52.6 ± 3.4 ^b^	129 ± 8 ^A^	2.7 ± 0.1 ^a^	113 ± 3 ^A^	2.6 ± 0.1 ^a^	120 ± 3 ^A^	2.5 ± 0.1 ^a^	126 ± 3 ^A^	2.5 ± 0.1 ^a^	132 ± 4 ^A^
18:3n-3	1.1 ± 0.2 ^a^	70 ± 16 ^A^	6.0 ± 0.4 ^b^	15 ± 1 ^B^	1.3 ± 0.0 ^a^	56 ± 1 ^A^	1.3 ± 0.0 ^a^	59 ± 1 ^A^	1.2 ± 0.0 ^a^	63 ± 2 ^A^	1.2 ± 0.1 ^a^	65 ± 3 ^A^
18:4n-3	1.3 ± 0.1 ^a^	84 ± 6 ^A^	---	---	1.2 ± 0.1 ^a^	51 ± 3 ^C^	1.3 ± 0.0 ^a^	59 ± 1 ^BC^	1.2 ± 0.0 ^a^	60 ± 2 ^BC^	1.2 ± 0.1 ^a^	63 ± 4 ^B^
20:2n-6	0.5 ± 0.0 ^a^	34 ± 2 ^A^	---	---	0.5 ± 0.0 ^a^	20 ± 0 ^D^	0.5 ± 0.0 ^a^	22 ± 1 ^CD^	0.5 ± 0.0 ^a^	25 ± 1 ^BC^	0.5 ± 0.0 ^a^	26 ± 2 ^B^
20:4n-6	2.4 ± 0.1 ^a^	159 ± 7 ^A^	---	---	2.4 ± 0.1 ^a^	101 ± 6 ^C^	2.3 ± 0.0 ^a^	105 ± 1 ^C^	2.4 ± 0.1 ^a^	126 ± 3 ^B^	2.4 ± 0.1 ^a^	126 ± 6 ^B^
20:4n-3	0.7 ± 0.1 ^a^	47 ± 3 ^A^	---	---	0.6 ± 0.0 ^ab^	26 ± 1 ^C^	0.6 ± 0.0 ^b^	28 ± 1 ^BC^	0.6 ± 0.1 ^b^	29 ± 4 ^BC^	0.6 ± 0.0 ^b^	32 ± 2 ^B^
20:5n-3	10.3 ± 0.6 ^a^	681 ± 42 ^A^	---	---	9.8 ± 0.2 ^ab^	411 ± 7 ^D^	9.3 ± 0.1 ^b^	435 ± 4 ^CD^	9.6 ± 0.1 ^ab^	494 ± 7 ^BC^	9.7 ± 0.4 ^ab^	513 ± 20 ^B^
21:5n-3	0.3 ± 0.0 ^a^	21 ± 1 ^A^	---	---	0.3 ± 0.0 ^a^	14 ± 0 ^C^	0.3 ± 0.0 ^a^	15 ± 0 ^C^	0.4 ± 0.1 ^b^	20 ± 3 ^AB^	0.3 ± 0.0 ^ab^	18 ± 1 ^B^
22:4n-6	0.5 ± 0.1 ^a^	30 ± 7 ^A^	---	---	0.3 ± 0.0 ^ab^	14 ± 1 ^B^	0.3 ± 0.0 ^b^	14 ± 0 ^B^	0.4 ± 0.1 ^ab^	18 ± 3 ^B^	0.3 ± 0.0 ^ab^	17 ± 1 ^B^
22:5n-6	1.0 ± 0.1 ^a^	63 ± 5 ^A^	---	---	1.0 ± 0.1 ^a^	40 ± 3 ^C^	0.9 ± 0.0 ^a^	40 ± 1 ^C^	1.0 ± 0.0 ^a^	52 ± 0 ^B^	1.0 ± 0.1 ^a^	50 ± 3 ^B^
22:5n-3	2.7 ± 0.2 ^a^	177 ± 15 ^A^	---	---	2.7 ± 0.1 ^a^	111 ± 4 ^C^	2.5 ± 0.0 ^a^	116 ± 2 ^C^	2.9 ± 0.1 ^a^	147 ± 5 ^B^	2.7 ± 0.2 ^a^	143 ± 11 ^B^
22:6n-3	29.7 ± 2.3 ^a^	1957 ± 154 ^A^	---	---	29.3 ± 2.2 ^a^	1227 ± 92 ^D^	26.4 ± 0.3 ^a^	1234 ± 14 ^CD^	30.0 ± 0.4 ^a^	1544 ± 20 ^BC^	29.3 ± 2.3 ^a^	1552 ± 121 ^B^
ΣPUFA	54.0 ± 3.1 ^ab^	3565 ± 202 ^A^	58.6 ± 3.8 ^a^	144 ± 9 ^D^	53.2 ± 2.5 ^ab^	2229 ± 105 ^C^	49.4 ± 0.5 ^b^	2307 ± 22 ^C^	53.9 ± 0.3 ^ab^	2777 ± 17 ^B^	52.9 ± 3.0 ^ab^	2805 ± 158 ^B^
Σn-3 PUFA	46.4 ± 3.1 ^a^	3060 ± 206 ^A^	6.0 ± 0.4 ^b^	14.9 ± 1.0 ^D^	45.5 ± 2.4 ^a^	1909 ± 99 ^C^	42.0 ± 0.4 ^a^	1960 ± 20 ^C^	46.1 ± 0.3 ^a^	2374 ± 17 ^B^	45.4 ± 2.8 ^a^	2489 ± 43 ^B^
Σn-6 PUFA	6.7 ± 0.7 ^a^	441 ± 47 ^A^	52.6 ± 3.4 ^b^	129 ± 8 ^D^	6.9 ± 0.2 ^a^	291 ± 7 ^C^	6.6 ± 0.1 ^a^	307 ± 5 ^BC^	6.9 ± 0.0 ^a^	354 ± 1 ^BC^	6.8 ± 0.2 ^a^	365 ± 4 ^B^
Σn-3/Σn-6	7.0 ± 0.9 ^a^ ± ^a^		0.1 ± 0.0 ^b^ ± ^a^		6.6 ± 0.2 ^a^ ± ^a^		6.4 ± 0.1 ^a^ ± ^a^		6.7 ± 0.1 ^a^ ± ^a^		6.7 ± 0.2 ^a^ ± ^a^	
IA	0.2 ± 0.0 ^ab^ ± ^a^		0.1 ± 0.0 ^c^ ± ^a^		0.2 ± 0.0 ^ab^ ± ^a^		0.3 ± 0.0 ^a^ ± ^a^		0.2 ± 0.0 ^b^ ± ^a^		0.2 ± 0.0 ^b^ ± ^a^	
IT	0.1 ± 0.0 ^ab^ ± ^a^		0.2 ± 0.0 ^a^ ± ^a^		0.1 ± 0.0 ^ab^ ± ^a^		0.2 ± 0.0 ^a^ ± ^a^		0.1 ± 0.0 ^b^ ± ^a^		0.1 ± 0.0 ^b^ ± ^a^	

Values are presented as mean ± standard deviation. Different lowercase letters on the same row correspond to statistical differences between the relative fatty acid profiles of each sample (*p* < 0.05). Different uppercase letters on the same row correspond to statistical differences between the absolute fatty acid profiles of each sample (*p* < 0.05). IA—index of atherogenicity; IT—index of thrombogenicity.

**Table 5 foods-13-01332-t005:** Bioaccessible fatty acid profiles (in mg/100 g ww) and percent (%) bioaccessibility of total lipids and each fatty acid for chub mackerel and hamburgers (raw and cooked).

	Raw Chub Mackerel		Raw Hamburger		Steamed Hamburger		Roasted Hamburger		Grilled Hamburger	
	(g/100 g)	% Bioac.	(g/100 g)	% Bioac.	(g/100 g)	% Bioac.	(g/100 g)	% Bioac.	(g/100 g)	% Bioac.
Total Lipids	5.3 ± 1.0 ^a^	73 ± 8 ^A^	4.9 ± 0.6 ^a^	95 ± 5 ^B^	5.4 ± 0.2 ^a^	96 ± 3 ^B^	5.4 ± 0.3 ^a^	84 ± 5 ^AB^	5.8 ± 0.3 ^a^	87 ± 1 ^AB^
Fatty Acid	(mg/100 g)	% Bioac.	(mg/100 g)	% Bioac.	(mg/100 g)	% Bioac.	(mg/100 g)	% Bioac.	(mg/100 g)	% Bioac.
16:0	731 ± 74 ^ab^	93 ± 9 ^A^	539 ± 43 ^b^	113 ± 7 ^AB^	743 ± 18 ^a^	117 ± 2 ^AB^	691 ± 32 ^ab^	116 ± 5 ^AB^	752 ± 90 ^a^	130 ± 15 ^B^
17:0	43 ± 4 ^a^	91 ± 8 ^A^	34 ± 1 ^b^	108 ± 2 ^BC^	40 ± 1 ^ab^	108 ± 2 ^BC^	36 ± 1 ^ab^	93 ± 3 ^AB^	42 ± 2 ^ab^	117 ± 6 ^C^
18:0	258 ± 13 ^a^	63 ± 3 ^A^	197 ± 4 ^c^	72 ± 2 ^B^	241 ± 3 ^b^	80 ± 1 ^C^	196 ± 1 ^c^	60 ± 0 ^A^	268 ± 4 ^a^	87 ± 1 ^D^
17:1	24 ± 2 ^b^	76 ± 6 ^A^	21 ± 1 ^a^	99 ± 3 ^B^	26 ± 0 ^bc^	110 ± 1 ^C^	23 ± 1 ^ab^	96 ± 3 ^B^	28 ± 1 ^c^	113 ± 3 ^C^
18:1n-9	646 ± 24 ^a^	79 ± 3 ^A^	530 ± 36 ^b^	104 ± 5 ^BC^	647 ± 1 ^a^	101 ± 0 ^BC^	651 ± 1 ^a^	96 ± 0 ^B^	695 ± 15 ^a^	108 ± 2 ^C^
18:1n-7	156 ± 5 ^a^	76 ± 2 ^A^	125 ± 4 ^c^	100 ± 3 ^C^	144 ± 1 ^b^	98 ± 1 ^C^	143 ± 1 ^b^	91 ± 1 ^B^	158 ± 1 ^a^	103 ± 1 ^C^
20:1n-9	15 ± 1 ^bc^	14 ± 1 ^A^	11 ± 1 ^a^	18 ± 1 ^B^	14 ± 0 ^abc^	18 ± 0 ^B^	13 ± 1 ^ab^	15 ± 1 ^A^	16 ± 1 ^c^	18 ± 1 ^B^
18:2n-6	7 ± 2 ^a^	5 ± 1 ^A^	16 ± 0 ^d^	14 ± 0 ^C^	10 ± 0 ^bc^	8 ± 0 ^B^	8 ± 0 ^ab^	7 ± 0 ^AB^	11 ± 0 ^c^	13 ± 0 ^C^
18:3n-3	43 ± 10 ^a^	62 ± 12 ^A^	43 ± 1 ^ab^	78 ± 1 ^AB^	50 ± 1 ^ab^	85 ± 1 ^BC^	50 ± 1 ^ab^	79 ± 2 ^AB^	55 ± 3 ^b^	86 ± 5 ^C^
18:4n-3	48 ± 3 ^a^	57 ± 4 ^A^	49 ± 3 ^ab^	100 ± 6 ^B^	59 ± 0 ^c^	100 ± 1 ^B^	57 ± 2 ^bc^	95 ± 3 ^B^	64 ± 4 ^c^	100 ± 3 ^B^
20:2n-6	22 ± 1 ^bc^	65 ± 4 ^A^	18 ± 0 ^a^	91 ± 0 ^B^	20 ± 0 ^ab^	94 ± 2 ^B^	19 ± 0 ^ab^	75 ± 2 ^A^	24 ± 2 ^c^	91 ± 6 ^B^
20:4n-6	47 ± 2 ^a^	30 ± 1 ^A^	50 ± 3 ^a^	48 ± 3 ^C^	46 ± 0 ^a^	44 ± 0 ^C^	45 ± 1 ^a^	36 ± 1 ^B^	55 ± 3 ^b^	44 ± 2 ^C^
20:4n-3	23 ± 2 ^a^	50 ± 3 ^A^	22 ± 1 ^a^	82 ± 3 ^B^	24 ± 1 ^ab^	87 ± 2 ^B^	24 ± 3 ^a^	82 ± 10 ^B^	28 ± 2 ^b^	86 ± 5 ^B^
20:5n-3	323 ± 20 ^a^	47 ± 3 ^A^	336 ± 6 ^ab^	81 ± 1 ^BC^	374 ± 3 ^c^	86 ± 1 ^C^	365 ± 5 ^bc^	74 ± 1 ^B^	441 ± 17 ^d^	82 ± 3 ^C^
21:5n-3	12 ± 0 ^a^	59 ± 2 ^A^	13 ± 0 ^a^	94 ± 1 ^B^	14 ± 0 ^a^	98 ± 2 ^B^	14 ± 2 ^a^	72 ± 10 ^A^	18 ± 1 ^b^	91 ± 6 ^B^
22:4n-6	13 ± 3 ^a^	44 ± 9 ^A^	11 ± 1 ^a^	74 ± 5 ^B^	11 ± 0 ^a^	75 ± 1 ^B^	11 ± 2 ^a^	60 ± 9 ^AB^	13 ± 1 ^a^	76 ± 0 ^B^
22:5n-6	31 ± 2 ^a^	50 ± 4 ^A^	32 ± 2 ^a^	76 ± 6 ^C^	33 ± 0 ^a^	82 ± 1 ^C^	33 ± 0 ^a^	63 ± 0 ^B^	42 ± 3 ^b^	73 ± 1 ^BC^
22:5n-3	95 ± 8 ^a^	53 ± 4 ^A^	102 ± 3 ^a^	88 ± 3 ^C^	106 ± 2 ^a^	91 ± 1 ^C^	107 ± 4 ^a^	73 ± 3 ^B^	131 ± 10 ^b^	81 ± 4 ^BC^
22:6n-3	809 ± 64 ^a^	41 ± 3 ^A^	938 ± 66 ^ab^	72 ± 5 ^BC^	969 ± 11 ^b^	79 ± 1 ^C^	961 ± 13 ^ab^	62 ± 1 ^B^	1219 ± 95 ^c^	69 ± 2 ^BC^

Values are presented as mean ± standard deviation. Different lowercase letters on the same row correspond to statistical differences between the absolute bioacessible fatty acid content of each sample (*p* < 0.05). Different uppercase letters on the same row correspond to statistical differences between the lipid and fatty acid bioaccessibility percentages of the samples (*p* < 0.05).

**Table 6 foods-13-01332-t006:** Selenium and iodine contents (expressed in µg/100 g ww or, in the case of seaweed, µg/100 g dw) in the initial and bioaccessible samples of the ingredients (chub mackerel, boiled quinoa, and the seaweed *Saccorhiza polyschides*) and hamburgers (raw and cooked), and respective elemental bioaccessibility percentages.

Element		Ingredient			Hamburger			
		Raw Chub Mackerel	Quinoa (Boiled)	Seaweed *Saccorhiza polyschides* (Freeze-Dried) *	Raw	Steamed	Roasted	Grilled
Se	Initial (µg/100 g)	59 ± 4 ^A^	1.1 ± 0.1 ^A^	116 ± 0	48 ± 2 ^aA^	50 ± 4 ^aA^	54 ± 3 ^abA^	61 ± 5 ^bA^
	Bioacc. (µg/100 g)	47 ± 3 ^B^	0.08 ± 0.01 ^B^	<LOQ	38 ± 1 ^aB^	41 ± 3 ^aB^	46 ± 4 ^aB^	43 ± 2 ^aB^
	Bioaacessibility (%)	80 ± 0 ^¥^	7 ± 0 ^§^	<LOQ	80 ± 10 ^¥^	82 ± 13 ^¥^	85 ± 5 ^¥^	70 ± 0 ^¥^
I	Initial (µg/100 g)	25 ± 1 ^A^	<LOQ	36,720 ± 603 ^A^	224 ± 7 ^aA^	221 ± 2 ^aA^	232 ± 16 ^abA^	255 ± 6 ^bA^
	Bioacc. (µg/100 g)	19 ± 1 ^B^	ND	17,586 ± 289 ^B^	127 ± 4 ^aB^	153 ± 1 ^bB^	163 ± 11 ^bB^	153 ± 4 ^bB^
	Bioaacessibility (%)	78 ± 0 ^¥^	ND	48 ± 3 ^£^	57 ± 1 ^Ϫ^	69 ± 2 ^§^	70 ± 1 ^§^	60 ± 2 ^Ϫ^

Values are presented as mean ± standard deviation. Different lowercase letters in a row correspond to significant differences (*p* < 0.05) between the levels of elements in the different analyzed hamburgers (either initial or bioaccessible). For each element, different uppercase letters in a column correspond to significant differences (*p* < 0.05) between levels of elements in the initial sample and bioaccessible fraction. Different symbols (¥, §, Ϫ, and £) in a row correspond to significant differences (*p* < 0.05) between the percentage (%) of bioaccessible element in the different analyzed samples (ingredients and hamburgers). ND—not determined; LOQ: limit of quantification (I < 4.9 µg/100 g); * data published by [12].

**Table 7 foods-13-01332-t007:** Polyphenol content (expressed in mg GAE/g ww or, in the case of seaweed, mg GAE/g dw) and FRAP antioxidant activity (expressed in μmol Fe^2+^ equivalent/g ww or, in the case of seaweed, μmol Fe^2+^ equivalent/g dw) of the seaweed *Saccorhiza polyschides* and hamburgers (raw and cooked).

Property		Ingredient	Hamburger			
		Seaweed *Saccorhiza polyschides* (Freeze-Dried)	Raw	Steamed	Roasted	Grilled
Polyphenol content	(mg GAE/g)	1.25 ± 0.06	0.06 ± 0.02 ^b^	0.08 ± 0.01 ^ab^	0.12 ± 0.03 ^a^	0.02 ± 0.02 ^c^
FRAP	(µmol Fe^2+^ eq./g)	9.5 ± 0.8	1.1 ± 0.2 ^ab^	0.9 ± 0.3 ^b^	1.5 ± 0.5 ^a^	0.4 ± 0.3 ^b^

Values are presented as mean ± standard deviation. Different lowercase letters in a row correspond to significant differences (*p* < 0.05) between the levels of polyphenols/antioxidant activity in the different analyzed hamburgers.

## Data Availability

The original contributions presented in the study are included in the article, further inquiries can be directed to the corresponding author.

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
