# Peer review of "Mackerel and Seaweed Burger as a Functional Product for Brain and Cognitive Aging Prevention"

_foods, 2024, doi:10.3390/foods13091332_

Round 1
Reviewer 1 Report
Comments and Suggestions for Authors
The aim of the manuscript "Functional Products for Alzheimer's Disease and Cognitive Ageing Prevention: Mackerel and Seaweed Burger as a Case Study" is to develop and optimize a functional hamburger enriched with nutrients known to protect against cognitive aging, such as n-3 polyunsaturated fatty acids (PUFA) like docosahexaenoic acid (DHA), iodine (I), and other essential nutrients. Specifically, the study aims to utilize the richness of DHA in chub mackerel, high vitamin B9 levels in quinoa, and abundant iodine in the seaweed Saccorhiza polyschides to create a nutritional product. The manuscript also investigates the effects of different culinary treatments (steaming, roasting, grilling, vs raw) and digestion on the bioaccessibility of these nutrients. Additionally, the aim is to assess the potential of the developed hamburgers to meet daily requirements for EPA+DHA, selenium (Se), and iodine, considering their bioaccessibility levels.
The manuscript is interesting and fits well with the scope of the Journal. It is well-prepared in general. My specific comments are given below.
The abstract could benefit from providing more context regarding the significance of the research. For instance, it could briefly mention the prevalence of cognitive aging and Alzheimer's disease, highlighting the need for dietary interventions to support cognitive health.
While the abstract mentions the bioaccessibility levels of selenium and iodine, it does not specify whether these levels are considered high or low in relation to recommended dietary intakes.
The introduction is informative.
Materials and methods should be given in more detail, especially methods for lipid, carbohydrate, and protein determination. They are cited but should be given in brief for the sake of completeness.
The authors should provide a photograph of the burger.
The results are generally good and well-presented, but there is a trouble with the context. The authors should discuss why they are making the burger rather than a supplement in the form of a capsule. They should describe their motivation and novelty in detail.
The title needs to be changed since there is no evidence that presented burgers add to AD prevention. I suggest “Mackerel and Seaweed Burger as a Functional Products for Brain and Cognitive Ageing Prevention:”
Comments on the Quality of English LanguageMinor changes are required.
Author Response
Dear Sir,
The whole paper was subjected to a new assessment in order to address the reviewer (and editor) concerns. For your advantage, we send you the revised manuscript with all alterations properly highlighted.
Reviewer #1:
The aim of the manuscript "Functional Products for Alzheimer's Disease and Cognitive Ageing Prevention: Mackerel and Seaweed Burger as a Case Study" is to develop and optimize a functional hamburger enriched with nutrients known to protect against cognitive aging, such as n-3 polyunsaturated fatty acids (PUFA) like docosahexaenoic acid (DHA), iodine (I), and other essential nutrients. Specifically, the study aims to utilize the richness of DHA in chub mackerel, high vitamin B9 levels in quinoa, and abundant iodine in the seaweed Saccorhiza polyschides to create a nutritional product. The manuscript also investigates the effects of different culinary treatments (steaming, roasting, grilling, vs raw) and digestion on the bioaccessibility of these nutrients. Additionally, the aim is to assess the potential of the developed hamburgers to meet daily requirements for EPA+DHA, selenium (Se), and iodine, considering their bioaccessibility levels.
The manuscript is interesting and fits well with the scope of the Journal. It is well-prepared in general. My specific comments are given below.
Thank you for your positive appraisal.
The abstract could benefit from providing more context regarding the significance of the research. For instance, it could briefly mention the prevalence of cognitive aging and Alzheimer's disease, highlighting the need for dietary interventions to support cognitive health.
We are grateful for this idea and we tried to give this perspective within the abstract section.
While the abstract mentions the bioaccessibility levels of selenium and iodine, it does not specify whether these levels are considered high or low in relation to recommended dietary intakes.
These are high values, thus Se and I bioaccessibility is not the problem, but the overall intake of these elements by many populations. This aspect was highlighted in the text.
The introduction is informative.
Thank you for your positive appraisal.
Materials and methods should be given in more detail, especially methods for lipid, carbohydrate, and protein determination. They are cited but should be given in brief for the sake of completeness.
These methodologies were described in more detail in the revised version of the manuscript.
The authors should provide a photograph of the burger.
Photos of all types of raw and cooked burgers were provided (a main figure instead of a supplementary material).
The results are generally good and well-presented, but there is a trouble with the context. The authors should discuss why they are making the burger rather than a supplement in the form of a capsule. They should describe their motivation and novelty in detail.
This justification for this particular option was justified in the final of the Introduction section and at the beginning of the R&D. Basically, this burger met three objectives, that were advantageous with respect to a supplement: (i) provided the neuroprotective ingredients embedded in ingredients naturally rich in them, thus circumventing problematic extraction/refining processes normally involved in the fabrication of a supplement; (ii) ensured that substances with a diverse chemistry were better joined together, preventing, for instance, that lipophilic and hydrophilic molecules were held together by an emulsifying agent; and (iii) provided a more ‘natural’ and traditional food, which would be better accepted by an elderly population.
The title needs to be changed since there is no evidence that presented burgers add to AD prevention. I suggest “Mackerel and Seaweed Burger as a Functional Products for Brain and Cognitive Ageing Prevention:”
We agree with you and the title was changed accordingly.
We sincerely hope that the performed alterations are clear and acceptable. However, if needed, we will gladly take the changes in the manuscript even further.
With our best regards,
The Authors
Reviewer 2 Report
Comments and Suggestions for Authors
The Authors of the article developed functional burgers based on mackerel and quinoa with seaweed, onion, garlic and herbal spices. The effects of cooking processing (steaming, baking, grilling vs. raw) and digestion on bioavailability were evaluated. From a nutritional point of view, the burgers were high in n-3 PUFAs and low in n-6 PUFAs, resulting in a high n-3/n-6 ratio and may meet the daily human requirement for EPA + DHA. This functional hamburger may furthermore be a suitable source of Se and I in the human diet, potentially providing biologically active and bioavailable nutrients to prevent cognitive aging.
The article is well written and contains interesting research, however, I have some comments on the scope of the research.
Specific Comments:
- The Authors write that quinoa is high in vitamin B9. So why didn't they examine the levels of this vitamin in quinoa-enriched hamburgers?
- Given the content of plant components rich in polyphenols, it would be worthwhile to examine the levels of these compounds.
- And finally, I think it is necessary to determine the antioxidant activity of hamburgers (to prove another health-promoting aspect of the product), due to the unsaturated fatty acid content and the large addition of plants.
Author Response
Dear Sir,
The whole paper was subjected to a new assessment in order to address the reviewer (and editor) concerns. For your advantage, we send you the revised manuscript with all alterations properly highlighted.
Reviewer #2:
The Authors of the article developed functional burgers based on mackerel and quinoa with seaweed, onion, garlic and herbal spices. The effects of cooking processing (steaming, baking, grilling vs. raw) and digestion on bioavailability were evaluated. From a nutritional point of view, the burgers were high in n-3 PUFAs and low in n-6 PUFAs, resulting in a high n-3/n-6 ratio and may meet the daily human requirement for EPA + DHA. This functional hamburger may furthermore be a suitable source of Se and I in the human diet, potentially providing biologically active and bioavailable nutrients to prevent cognitive aging.
This is precisely the concept and idea behind this paper in a nutshell.
The article is well written and contains interesting research, however, I have some comments on the scope of the research.
Thank you for your positive appraisal and we will try to improve the manuscript according to your criticisms and suggestions.
Specific Comments:
- The Authors write that quinoa is high in vitamin B9. So why didn't they examine the levels of this vitamin in quinoa-enriched hamburgers?
That is the third part of our study that aims to address the effect of thermal treatments (such as in grilled burgers) on particularly thermally sensitive vitamins (as B9, B12, and vitamin D). We have already published a paper on the storage stability of this hamburger (this should be the second part, but it was already published). This is the reference:
Valentim, J., Afonso, C., Gomes, R., Gomes-Bispo, A., Prates, J.A.M., Bandarra, N.M. and Cardoso, C. 2024. Influence of cooking methods and storage time on colour, texture, and fatty acid profile of a novel fish burger for the prevention of cognitive decline. Heliyon, 10(5): e27171.
So, this is a multi-part and multi-manuscript experimental work corresponding to an important research project.
- Given the content of plant components rich in polyphenols, it would be worthwhile to examine the levels of these compounds.
You are right. In particular, seaweeds, particularly brown seaweeds, are rich in polyphenols and we did determine them (other vegetable ingredients may also contribute, despite their very small incorporation percentages). We have added a table and these results as well as the correlative discussion of them.
- And finally, I think it is necessary to determine the antioxidant activity of hamburgers (to prove another health-promoting aspect of the product), due to the unsaturated fatty acid content and the large addition of plants.
We have also these results and can also include them here. Our main doubt was rendering the manuscript too lengthy. However, we will keep it focused and synthetic.
We sincerely hope that the performed alterations are clear and acceptable. However, if needed, we will gladly take the changes in the manuscript even further.
With our best regards,
The Authors
Round 2
Reviewer 1 Report
Comments and Suggestions for Authors
The authors addressed all my comments.
Comments on the Quality of English LanguageMinor changes are required.